# Primary Care and Physical Literacy: A Non-Randomized Controlled Pilot Study to Combat the High Prevalence of Physically Inactive Adults in Austria

**DOI:** 10.3390/ijerph18168593

**Published:** 2021-08-14

**Authors:** Peter Holler, Johannes Jaunig, Othmar Moser, Silvia Tuttner, Helmut Simi, Dietmar Wallner, Frank Michael Amort, Mireille van Poppel

**Affiliations:** 1Institute of Health Management in Tourism, FH JOANNEUM—University of Applied Sciences, 8344 Bad Gleichenberg, Austria; silvia.tuttner@fh-joanneum.at (S.T.); helmut.simi@fh-joanneum.at (H.S.); dietmar.wallner@fh-joanneum.at (D.W.); frank.amort@fh-joanneum.at (F.M.A.); 2Sport Science Laboratory, FH JOANNEUM—University of Applied Sciences, 8344 Bad Gleichenberg, Austria; 3Institute of Human Movement Science, Sport and Health, University of Graz, 8010 Graz, Austria; johannes.jaunig@uni-graz.at (J.J.); mireille.van-poppel@uni-graz.at (M.v.P.); 4Division Exercise Physiology and Metabolism, Department of Sport Science, University of Bayreuth, 95447 Bayreuth, Germany; othmar.moser@uni-bayreuth.de; 5Division of Endocrinology and Diabetology, Department of Internal Medicine, Medical University of Graz, 8010 Graz, Austria

**Keywords:** physical literacy, primary care, intervention, measurement, physical inactivity, physical activity, adults

## Abstract

The multidimensional concept of physical literacy is fundamental for lifelong physical activity engagement. However, physical literacy-based interventions are in their infancy, especially among adults. Therefore, the purpose of this pilot study was to assess the association of a physical literacy-based intervention with changes in self-reported physical literacy among inactive adults. A non-randomized controlled study (2 × 2 design) was conducted, comparing pre- vs. postintervention. Twenty-eight inactive healthy participants in the intervention group (89% female, 53 ± 10 years) entered a physical literacy-based intervention once weekly for 14 weeks. The non-treated control group consisted of 22 inactive adults (96% female, 50 ± 11 years). Physical literacy was evaluated with a questionnaire encompassing five domains: physical activity behavior, attitude/understanding, motivation, knowledge, and self-efficacy/confidence. ANOVA models were applied to evaluate changes by time and condition. Following the intervention, significant improvements were seen for overall physical literacy and in four out of five physical literacy domains, including physical activity behavior, attitude/understanding, knowledge, and self-efficacy/confidence (all *p* < 0.01, Cohen’s *d* = 0.38–0.83). No changes by time x condition were found for motivation. The physical literacy-based intervention applied in this study may be a promising approach to help inactive adults to adopt an active lifestyle.

**Trial registration:** German Clinical Trials Register (DRKS), DRKS00024690, date of registration: 13.04.2021, retrospectively registered.

## 1. Introduction

There is a compelling body of literature highlighting the physiological [1,2] and psychological benefits [3,4] of being sufficiently physically active. However, a physically inactive lifestyle has become the norm in many developed countries over the last decades [5], with a similar trend observed in Austria. The two latest Austrian Health Interview Surveys, conducted in 2014 and 2019, both found that more than 75% of all Austrian adults do not meet national recommendations for aerobic and strengthening physical activity [6,7], i.e., at least 150 min of moderate to vigorous aerobic activity per week and muscle-strengthening activity on two or more days per week [8]. By not doing so, most of the Austrian population are putting themselves at a higher risk of negative health outcomes. In fact, globally, as well as in Austria, 6 to 10% of the burden of disease from coronary heart disease, type 2 diabetes, breast, and colon cancer are directly associated with physical inactivity [1]. Therefore, implementing effective and sustainable interventions to increase the physical activity of inactive individuals is crucial.

Over the last several years understanding the factors influencing physical activity has shifted from the individual-level factors to a socio-ecological model assuming that physical activity participation is affected by intrapersonal, interpersonal, environmental, and political factors [9,10]. Although this approach provides a more comprehensive and accurate understanding of physical activity, the focus on individual-level factors is still important. A great amount of responsibility for one’s physical activity level lies with the individual and, thus, effective interventions at the individual level remain of particular interest to researchers and practitioners [11].

Over the past few years, the concept of physical literacy has attracted increasing attention. While evidence from prospective studies is still lacking [12], the concept of physical literacy assumes that a certain set of intrapersonal factors is required to achieve and perpetuate a long-term physically active lifestyle [13,14]. There are many definitions of physical literacy [15,16], but most of the previously published studies used the physical literacy definition established by Margaret Whitehead [12]. As a holistic framework, she describes physical literacy “…as the motivation, confidence, physical competence, knowledge and understanding to value and take responsibility for engagement in physical activities for life” [17] (p. 11). In its essence, physical literacy is most accurately summarized as one’s capacity for and commitment to a physically active lifestyle [18]. By unifying several theory-based physical activity determinants of well-established psychological behavior change theories (e.g., social cognitive theory or self-determination theory [19,20]), the concept of physical literacy is often considered as the gateway to lifelong physical activity engagement [21], but it may also provide a theoretical framework for effective and sustainable interventions to increase long-term physical activity [22]. Furthermore, in light of its link to lifelong physical activity participation, physical literacy has been conceptualized as an essential determinant of physical, mental, and social health [23]. The importance of physical literacy is highlighted in national and international policy documents on physical activity promotion, encompassing the working group report on the Austrian Health Target 8 [24] and the World Health Organization’s Global Action Plan on physical activity [25].

Despite the increasing emphasis on physical literacy, empirical evidence lags behind the theoretical development of the concept. Since only a few studies have addressed the construct of physical literacy within the context of health [12], evidence for the relationship between physical literacy and health indicators (e.g., cardiorespiratory fitness, CRF) is still needed, particularly in adulthood. Moreover, it was proposed that all individuals can develop their physical literacy regardless of age, ability, weight, and height [14,15,26]. However, only few studies have addressed physical literacy-based intervention so far, with most of them focusing on children and adolescents [12,27]. The adult population has largely been ignored. This is an important gap, since physically inactive adults in particular are suggested to benefit from intervention targeting physical literacy [28,29]. Physical literacy is a dynamic concept and conceptualized as a lifelong journey, rather than a certain endpoint that, once reached, lasts for the rest of one’s life [30,31]. Consequently, one’s physical literacy might vary and be characterized by successes and setbacks over time, depending on a person’s experience with physical activity [28,29]. Since physically inactive adults often report negative experiences of physical activity in their past, it is likely that these individuals especially have had a stagnant or low-progress physical literacy journey with many setbacks [28].

For it to be effective, several principles must be considered when designing and implementing a physical literacy-based intervention for physically inactive adults. Physical literacy can be operationalized as a multidimensional construct with underlying interconnected domains [15]. We conceptualized the domains into five subscales, which were adopted but slightly changed from the physical literacy definition given above: (i) motivation; (ii) knowledge; (iii) self-efficacy/confidence; (iv) attitude/understanding; and (v) physical activity behavior (a more detailed description of the domains and intervention approach can be found in Appendix A: Detailed description of the domains and intervention approach ). In light of this, a physical literacy-based intervention must target all these domains simultaneously. Thereby, domain-specific strategies must be applied to elicit improvements in each domain, such as mastery experience or verbal persuasion as important sources of self-efficacy [20,32]. Clearly, the intervention must be anchored in a motivational, empowering, and enjoyable context, per se, to trigger a positive impact, particularly on the affective domains [23,33]. Moreover, since social support and relationships relate to affective elements of physical activity [19] and, consequently, to long-term physical activity maintenance [34], a physical literacy-based intervention must also incorporate social elements promoting interactions with peers as well as a cohesion among the group. Finally, apart from providing an effective intervention, the recruitment of inactive adults to participate in such an intervention is always a challenge, especially of inactive individuals with no previous exercise experience and existing chronic diseases [35,36]. It is estimated that up to 80% of adults in developed countries (with about 82% in Austria) visit their general practitioners at least once a year, especially chronically ill and inactive adults [37]. Therefore, a primary care setting can be very effective for reaching inactive individuals for participation in a physical literacy-based intervention [36]. Indeed, patients respect general practitioners as well as other health professionals and consider their advice a credible and important source of health-related information [38].

Between 2016 and 2017, the project “Primary Care und Physical Literacy” was developed in Austria. The aim of the project was to identify physically inactive adults at primary care level and to motivate them to participate in a physical literacy-based intervention, designed in compliance with all the described principles. At the heart of the project two non-randomized controlled studies (referred as previous study and present study) were conducted consecutively in two different rural regions of Styria, a province of Austria (data collecting phase, previous study: October 2016–February 2017; present study: March 2017–June 2017). Overall, following a pilot approach, both studies were small-scale and conducted not only to examine the associations of a physical literacy-based intervention with changes in physical literacy, but also to test the feasibility of the applied research protocols and the sample recruitment strategy within a primary care setting, in preparation for future large scale randomized controlled trials.

The results from the previous study have recently been published [39]. It was shown that participating in a low-dose physical literacy-based intervention (once weekly, over 15 weeks) is safe and feasible for inactive adults and associated with an increase of overall physical literacy. Although this is a very promising finding, the present study, as a second pilot study, was conducted to strengthen the reliability in terms of the observed effect size and generalizability of this previous finding [40]. Several shortcomings from the previous pilot study, regarding the physical literacy measurement tool and the corresponding scoring procedure, were improved in the present study. Besides, the recruitment strategy in our previous study was not exclusively linked to a primary care setting, but also included regional newspapers advertisements and a social media campaign. Therefore, no accurate conclusions could be drawn about the effectiveness of a primary care setting in the recruitment of physically inactive adults.

Compared to the previous study, the present study was conducted in a different region and at a different time of year and used a recruitment strategy tied exclusively to a primary care setting. Moreover, CRF was measured to provide evidence for the relationship between physical literacy and health indicators. The primary purpose of the present study was to assess the associations of a physical literacy-based intervention targeting physically inactive adults with changes in physical literacy. A secondary aim was to evaluate the associations between CRF, intervention parameters, sociodemographic, and anthropometric factors, and changes in physical literacy triggered by the intervention.

## 2. Materials and Methods

### 2.1. The Design, Participants, and Procedures

A non-randomized controlled parallel group study (intervention vs. non-treated control group) was conducted with outcome measurements at baseline and after a 14-week intervention period. Based on self-selection, study participants received either a physical literacy-based intervention (intervention group, referred as IG) or participated in a non-treated control group (CG). Eligibility criteria were identical to those in our previous study and were described there in detail [39]. Briefly, physically inactive adults aged between 18 and 65 years who received medical clearance to participate in the study were included. The study was approved by the Research Ethics Committee of the University of Graz, Styria, Austria (GZ. 39/16/63 ex 2016/17) and written informed consent was obtained from all participants prior to any study-related activities and measurement.

The recruitment period for both IG and CG lasted three weeks (1–21 March, 2017) and overlapped with the start of the physical literacy-based intervention on 14 March 2017. IG participants were recruited through a community health center and additionally through six general practitioners. The self-administered baseline measurement of physical literacy was supervised by the employees of the community health center and the general practitioners themselves and lasted about 15 min. In addition to the physical literacy assessment, sociodemographic and anthropometric data were collected and CRF (only in the IG) was measured. The baseline measurements of CRF were performed within one week of the start of the intervention, the follow-up measurement within one week of the end of the intervention. The recruitment of the CG and the corresponding measurements were conducted by the community health center. The staff was instructed to invite individuals into the control arm who were similar to the participants in the intervention group in terms of age, gender, and anthropometric parameters. Those individuals who were interested in the study but whose schedules did not allow them to participate in the intervention (e.g., because of work commitments) were specifically asked to participate in the control arm. CG participants were informed not to change their level of physical activity or daily habits during the duration of the study. This study was performed in Mürzzuschlag, a district of Styria, Austria, in cooperation with the FH JOANNEUM Bad Gleichenberg University of Applied Sciences (Styria, Austria).

### 2.2. Physical Literacy

Physical literacy was measured with a 55-item-questionnaire, which contained five subscales corresponding to our five domains of the physical literacy concept: (i) motivation; (ii) knowledge; (iii) self-efficacy/confidence; (iv) attitude/understanding; and (v) physical activity behavior. The applied questionnaire presented a slightly adapted version of the physical literacy questionnaire used in our previous study [39], with Cronbach’s alpha values ranging from 0.80 to 0.95 in the former subscales (domains). The adaptions were related to items in the self-efficacy/confidence and physical activity behavior domains because of the methodological shortcomings of these former subscales (described in detail later in this section). The English version of the questionnaire used in this study can be found in Appendix A: Physical Literacy Questionnaire (English version).

#### 2.2.1. Motivation, Knowledge, and Attitude/Understanding Domains

The measurement and scoring procedures of these domains were identical to those in the original questionnaire [33]. Briefly, motivation was measured with three subscales, adopted from the Behavioral Regulation in Exercise Questionnaire-2 (BREQ-2, [41]), Situational Motivation Scale (SIMS, [42]), and Sport Motivation Scale (SMS28, [43]). The subscales were merged into one motivation scale with 12 items in total. Knowledge was assessed with five open-ended items, based on a previous study by Fitzgerald et al. [44]. Attitude was measured with 10 items, four of them from the Stanford Five City Study questionnaire [45] and six constructed by the authors themselves to provide a more comprehensive assessment of this domain.

#### 2.2.2. Self-Efficacy/Confidence Domain

This domain was assessed by psychometric scales regarding exercise self-efficacy adopted from Renner and Schwarzer (21 items in total, rated on a 5-point Likert-type scale [46]). In the original questionnaire, the domain was additionally covered by the Body Image Questionnaire (FKB-20, [47]), as an assessment tool for physical self-confidence. Both measurement instruments were combined to generate a single domain score, and, consequently, the domain was previously designated as the “self-confidence & self-efficacy” domain. Apart from the fact that there was an uneconomically high number of items covering this single domain (i.e., 41 items), the use of two similar, yet different aspects within this domain hampered the interpretation of the overall domain score [14]. Moreover, the term “confidence” in Whitehead’s physical literacy represents a definition of physical self-efficacy rather than one of self-confidence (a rationale for this can be found in the Appendix A).

#### 2.2.3. Physical Activity Behavior Domain

Physical activity behavior was assessed with the International Physical Activity Questionnaire Short Form (IPAQ-SF). The IPAQ-SF is a widely used self-report tool to measure physical activity at different intensity levels over the previous seven days and has demonstrated adequate validity and reliability in previous research [48]. The domain score was calculated by summing up the total weekly minutes of moderate and vigorous physical activity, with a higher score indicating a greater physical activity behavior. Walking minutes were not considered for the score calculation.

#### 2.2.4. Physical Literacy Composite Score and Domain Scores

Domain scores were generated by averaging all items of a single domain (except for physical activity behavior), with higher values representing a greater domain proficiency. Thereafter, standardized scores using a mean of 100 and a standard deviation of 10 were generated separately for each of the five domain scores and averaged to yield a total physical literacy score, with higher values indicating a greater physical literacy. Since the observed variances among the five domain scores were markedly different in the present study (pre- and postintervention), the use of standardization to create a composite score was essential to avoid biased results from the heterogeneity of variances [49].

### 2.3. Cardiorespiratory Fitness (CRF)

Maximum oxygen uptake (VO_2max_) was assessed as an indicator of CRF. Thereby, VO_2max_ was predicted based on the performance in the Chester Step Test using the standard testing protocol [50,51]. Recent studies on the reliability and validity of this test have demonstrated high test–retest reliability [52] as well as a strong correlation for convergent validity (*r* = 0.92), when compared to a standardized cardiopulmonary exercise test until exhaustion among a large group of healthy adults [51].

### 2.4. Sociodemographic and Anthropometric Data

Self-reported age (years), body weight (kg), height (cm), body-mass index (BMI) (kg/m^2^), education level, and chronic diseases were collected in the physical literacy questionnaire.

### 2.5. The Physical Literacy-Based Intervention

The physical literacy-based intervention was similar to the one in our previous study [39] and differed mainly in terms of the organizational framework. The intervention period lasted 14 weeks, whereby the intervention was delivered once weekly (i.e., a total of 14 intervention days). All intervention days consisted of two consecutive 50-min-sessions, focusing either on strength- or on multimodal (combination of strength and aerobic)-related activities. Participants were not allocated to a specific type of session. In fact, they were allowed to switch between the different types of sessions each week as well as to attend all sessions on one intervention day. In keeping with the self-determination theory [19], the aim of this approach was to increase the participants’ adherence and enjoyment by providing them with more autonomy in terms of the intervention time and the type of session [53]. Overall, all sessions were carefully designed according to the principles described in the Introduction and led by an experienced sport scientist. Specifically, each session started with a five-minute warm-up walking game like “Chain trapping” or “Mirroring”, followed by a period of major joint mobilization. The subsequent main part of the session included session-specific exercises mixed with elements aimed at physical activity knowledge transfer. Indeed, a strong focus was placed on a physical activity knowledge transfer within the intervention setting as knowledge is considered not only an essential predictor of motivation [54,55], but also a positive influence on individuals’ attitudes toward physical activity [56,57]. The strength-based sessions included simple body weight exercises and exercises with various training tools (e.g., resistance bands). There was a strong focus on partner exercises to promote social interactions, which are a well-established source of motivation, according to the self-determination theory [19]. During the recovery periods (usually after five exercises with four periods per session, lasting three minutes) and also during the exercises, the instructor provided information not only about how to perform the exercises correctly, but also about the “everyday relevance” of each exercise and the health benefits of aerobic and strength exercises and regular physical activity in general. All messages to the participants were framed positively with a strong focus on the affective aspects of physical activity (i.e., the focus was on exercise experience and the benefits of physical activity rather than on the hazards of inactivity), since this approach is more effective at changing one’s attitude toward physical activity [58,59]. The main part of the multimodal-based sessions was separated into two parts. In the first part, a learning-oriented walking/running game was played that was designed to elicit physical activity knowledge transfer (one game per session; an example can be found in our previous study). The second part included strength-related exercises taken from the strength-based sessions (usually 10 exercises with two recovery periods per session). To ensure mastery experience, the most powerful influence on physical activity self-efficacy [20,53], all exercises and games in both sessions were tailored to participants’ physical and mental constitutions. At the end of each session, participants were given individual, positive, and encouraging feedback as an additional strategy to enhance physical activity self-efficacy. Additionally, print materials outlining the national physical activity guidelines or the benefits of being physically active, and worksheets focusing on physical activity goal setting, planning, and self-monitoring were distributed and discussed. The aim of these worksheets was to develop self-regulatory skills as a well-established intervention strategy in changing physical activity behavior [59]. Heart rate was monitored continuously during each session and participants’ rated perceived exertions (RPE scale, 6 to 20, [60]) were checked at the end of each session to ensure a moderate intensity level (i.e., heart rate: 64–76% of HR_max_; RPE scale: 12–13 [61]). HR_max_ was age-predicted using the following equation: HR_max_ = 208 − (0.7 * age) [62]. Adherence to the intervention was calculated by the ratio between attended intervention days and theoretically possible intervention days (given as percentage). A Template for Intervention Description and Replication (TIDieR) checklist regarding our intervention can be found in Appendix A: Template for Intervention Description and Replication (TIDieR) checklist. 

### 2.6. Sample Size

Based on the observed effect sizes for the total physical literacy score in our previous study [39], an a priori sample size calculation was performed using G*Power version 3.1.9.4 [63]. Thereby, the partial eta squared (ηp²) from the time × group interaction between the intervention group subsample recruited by general practitioners (IG-GP) and the control group (CG) was used as an effect estimate, as this previous result corresponded most closely to the present study protocol. To detect an effect of ηp² = 0.18 with a power of at least 80% in a mixed ANOVA model (2 groups, 2 measurement points, α = 0.05, two-tailed) a total sample size of *n* ≥ 40 was required. This target reflected the need to recruit at least 20 participants to each group (intervention and control group).

### 2.7. Statistical Analysis

Results are expressed as mean and standard deviation (M ± SD) for continuous or as frequency (%) for categorical variables, unless otherwise indicated. Normal distribution was checked for each continuous variable (skewness lower than ± 1.5). Independent *t*-tests, Mann–Whitney U-test or χ2-test were used to assess between-group comparisons at baseline. To test the reliability of the physical literacy questionnaire, Cronbach’s alpha coefficient (α) as an indicator of internal consistency was used, with α ≥ 0.7 indicating acceptable internal consistency [64]. To evaluate the magnitude of the treatment effect on the physical literacy composite score and domain scores, a series of mixed ANOVAs with the two groups (IG and CG) as between-subject factors and the outcome assessments (baseline and postintervention scores) as within-subject factors were applied. In cases of a significant ANOVA, Bonferroni-adjusted post hoc analyses were performed to evaluate within-group changes over time. Since no significant influence of any covariate on the primary outcome was found in a preceding regression analysis (data not shown), none were introduced in the ANOVA models. The effect sizes were calculated using the partial eta-square (ηp²), with ηp² ≥ 0.01 indicating a small, ηp² ≥  0.059 a moderate and ηp² ≥  0.14 a large effect, or Cohen’s *d* for comparison of mean values with *d* ≥ 0.2 indicating a small, *d* ≥ 0.5 moderate, and *d* ≥ 0.8 a large effect [65]. Moreover, a stepwise regression analysis was performed to identify factors influencing changes in physical literacy (i.e., physical literacy postintervention minus physical literacy baseline). Age, gender, pre- and postintervention values, and change values regarding BMI and VO_2max_, education levels, chronical diseases (yes/no), and intervention parameters including adherence, performed total sessions and minutes, as well as exercise intensity were introduced as predictors in the stepwise procedure. All statistical analyses were conducted using SPSS software (IBM Corp. Released 2017. IBM SPSS Statistics for Windows, Version 25.0. Armonk, NY, USA) and GraphPad Prism version 7 (GraphPad Software, La Jolla, CA, USA), with a two-tailed *p*-values < 0.05 indicating statistical significance.

## 3. Results

A total of 103 individuals were invited to participate in the study, of which 45 declined (response rate = 56%). Consequently, 58 participants were allocated by self-selection to either the IG or the CG. In the IG, 34 participants were screened for eligibility, of which 33 met inclusion criteria and received the physical literacy-based intervention. Five participants of the IG withdrew from the study during the first two weeks of the study due to health problems not related to the intervention (*n* = 2), work commitments that precluded ongoing participation (*n* = 1), and unknown reasons (*n*= 2). As a result, data of 28 IG participants were analyzed. Out of these 28 participants, a sub-sample of 23 (82%) additionally completed both baseline and follow-up measurements of CRF. A testing appointment prior to the first intervention session was not possible for the other six IG participants due to work commitments (*n* = 4) and holidays abroad (*n* = 2). In the CG, 24 participants were screened for eligibility, of which 22 met inclusion criteria and successfully completed the study. Figure 1 illustrates the flow diagram of recruitment numbers.

### 3.1. Baseline Characteristics

There were no significant differences in the baseline characteristics between participants in the IG and CG (all *p* > 0.05) (Table 1) or in the analysis of baseline characteristics of the CG and IG subsample who additionally performed CRF measurements (data not shown). At baseline, the IG subsample had a relative VO_2max_ of 30.3 ± 4.2 mL kg^−1^ min^−1^.

### 3.2. Adherence and Intervention Parameters

Since recruitment and intervention periods slightly overlapped, seven (25%) participants joined the intervention for the first time one week after the actual onset. Total average adherence to the intervention was 68 ± 20%. Of the 28 IG participants, twelve (43%) participated 3.2 ± 3.2 times in both the strength- and multimodal-based sessions on the same intervention day. IG participants performed a mean of 10.8 ± 3.7 sessions in total. The mean exercise intensity, as measured by heart rate in percent of participants, predicted HR_max_ and the RPE scale were 65 ± 6% and 12 ± 1, respectively. Both intensity parameters indicate a moderate intensity level [61]. No adverse events were reported during the intervention period.

### 3.3. Reliability of the Physical Literacy Questionnaire

For the analysis of Cronbach’s α coefficients, data from the IG and CG participants were analyzed jointly. The coefficients for the pre- and postintervention scale scores were 0.86 and 0.85 for the motivation domain, 0.78 and 0.83 for the attitude domain, and 0.93 and 0.96 for the self-efficacy/confidence domain [64]. Since the knowledge and physical activity behavior domains were represented by formative measurement models, Cronbach’s α coefficients were not calculated for these domains [66].

### 3.4. Physical Literacy

For the physical literacy composite score, results from the ANOVA showed a significant time × group interaction with a large effect size (*p* < 0.001, ηp² = 0.29). Subsequent post hoc tests indicated that physical literacy increased significantly among IG participants (*p* = 0.038, *d* = 0.38) over time, while participants in the CG showed a significant decrease (*p* = 0.002, *d* = −0.56; Figure 2a). Means and standard deviations by time and condition and corresponding test statistics for the physical literacy composite score as well as for all domain scores can be found in the Appendix B (see Table A1).

### 3.5. Domain Scores of Physical Literacy

#### 3.5.1. Physical Activity Behavior Domain

A statistically significant time × group interaction with a moderate effect size (*p* = 0.02, ηp² = 0.11) was found for physical activity behavior, after applying the ANOVA. Subsequent post hoc analysis revealed a significantly higher domain score among IG participants at postintervention in comparison to the baseline (*p* = 0.005, *d* = 0.77), while no detectable change was observed for the CG (*p* = 0.56, *d* = −0.15; Figure 2b).

#### 3.5.2. Attitude/Understanding Domain

A significant time × group interaction effect with a large effect size (*p* < 0.01, ηp² = 0.16) was observed with ANOVA when IG and CG were compared over time regarding the domain score for attitude. Subsequent post hoc analysis detected a significant increase among participants in the IG (*p* < 0.001, *d* = 0.52), whereas the score remained relatively consistent among CG participants over time (*p* = 0.87, *d* = −0.04; Figure 2c).

#### 3.5.3. Motivation Domain

Neither a significant main effect for time (*p* = 0.70, ηp² < 0.01) nor a significant time × group interaction (*p* = 0.44, ηp² = 0.01) was found regarding the motivation domain with the ANOVA (Figure 2d).

#### 3.5.4. Knowledge Domain

The ANOVA revealed a significant time × group interaction with a moderate effect size (*p* = 0.03, ηp² = 0.10) for the domain knowledge. Subsequent post hoc analysis showed that IG participants increased significantly over time (*p* = 0.004, *d* = 0.66), while there was no change among CG participants (*p* = 0.75, *d* = −0.08; Figure 2e).

#### 3.5.5. Self-Efficacy/Confidence Domain

Regarding the self-efficacy domain, results from the ANOVA showed a significant time × group interaction with a moderate effect size (*p* = 0.02, ηp² = 0.11). Subsequent post hoc tests indicated that physical literacy increased significantly among participants in the IG (*p* < 0.001, *d* = 0.83), while no detectable change was observed for the CG (*p* = 0.25, *d* = 0.19; Figure 2f).

### 3.6. Factors Influencing Changes in Physical Literacy

Because CRF measurements were available for only 23 out of 28 IG participants, two separate stepwise regression analyses were performed to identify factors influencing changes in physical literacy (i.e., physical literacy postintervention minus physical literacy baseline). The first model was run on data from all 28 IG participants, with VO_2max_ values not included as predictors. For the second model, VO_2max_ values were introduced as explanatory variables and, therefore, the analysis was only run on data for subjects from the IG subsample who additionally performed corresponding CRF measurements. Overall, only the second model including baseline VO_2max_ was proven to be adequate (R^2^ = 0.18), whereby baseline VO_2max_ was found to be negatively correlated with changes in physical literacy (β = −0.42, *p* = 0.04). No other variables were identified as significant predictors. Moreover, in an additional correlational analysis we found no association between baseline values of CRF and physical literacy (*r* = −0.12, *p* = 0.36).

### 3.7. Changes in Cardiorespiratory Fitness (CRF)

Even though improvements were observed in the hypothesized direction, no changes regarding VO_2max_ following the intervention condition were found in a paired *t*-test for the IG sub-sample completed CRF measurements (30.3 ± 4.2 vs. 31.4 ± 4.1 mL kg^−1^ min^−1^, *t*(22) = 1.5, *p* = 0.15, *d* = 0.28).

## 4. Discussion

As a gateway to lifelong physical activity engagement, the concept of physical literacy has attracted increasing attention over the past few years [13,14]. However, empirical evidence lags behind the theoretical development of the concept. To date, little effort has been devoted to physical literacy-based interventions, particularly among an adult population. Hence, we investigated the association of a physical literacy-based intervention targeting physically inactive adults with changes in physical literacy.

The principal finding of this study indicated that a 14-week physical literacy-based intervention, delivered once weekly, was able to elicit improvements with large effect sizes in overall physical literacy among physically inactive adults. This finding was supported by the fact that a very similar outcome was found in our previous study [39], which can be considered as a direct comparative study. Likewise, our results were in line with physical literacy-based intervention studies targeting different populations, such as first-year university students [67] or children aged between six and ten [68]. However, due to the nature of a pilot study we must be careful not to overgeneralize the significance of our findings. Drawing conclusions based on our findings should be done with caution.

In the evaluation of the specific physical literacy domains, we observed improvements with moderate to large effect sizes in four out of five domains. No changes were detected for the motivation domain following the intervention period. This was a somewhat surprising finding, since we not only applied a number of well-established strategies to enhance participants’ motivation within the intervention framework [53], but also found improvements in the knowledge domain, which is considered to be an essential predictor of motivation [54,55]. A possible explanation of this finding may be that participants who volunteered for the study might have been highly motivated at baseline and, consequently, no further improvements were found in this domain. Indeed, several studies acknowledged high baseline values for motivational variables in physical activity interventions because of the self-selection of participants [69,70]. Further studies on physical literacy should therefore consider and address these circumstances in study designs and recruitment procedures to prevent self-selection bias of participants.

Overall, the findings from our study may have important practical implications. Most participants in our study were initially physically inactive, overweight, and middle-aged women, who showed improvements not only in overall physical literacy but also in almost all physical literacy domains representing important theory-based determinants for the initiation and maintenance of regular physical activity. Our findings might be even more promising when considering the low dosage of our physical literacy-based intervention along with the fact that behavior changes are suggested to become more challenging with increasing age [71,72]. Furthermore, since physical literacy is positioned as the cornerstone of lifelong physical activity engagement [21], our observed improvements in physical literacy may have implications for the behavior not only in the short term but also in the long term. Not surprisingly, considering current exercise recommendations [61], the applied exercise dosage in our intervention might have been too low to improve participants’ CRF in this timeframe. However, given the hypothetical link between physical literacy and lifelong physical activity participation, it might be possible that further research incorporating follow-up measurements will find improvements in CRF over longer time periods.

As a secondary outcome of our study, we found that participants with lower CRF at baseline benefited more from the intervention in terms of the change in their overall physical literacy. However, we did not detect a correlation between physical literacy and CRF at baseline, which would have suggested that a lower baseline physical literacy in those participants with lower initial CRF was the underlying trigger of our finding. In the only study to date on this topic, Lang et al. [73] demonstrated contrary findings, namely, a clear association between CRF and overall/domain-specific physical literacy. This study included only school-aged children, however. Considering the small sample size along with the homogeneity of the obtained CRF levels in our study, we should not draw definitive conclusions about the cause of this outcome. Considerably more work is necessary to better understand the associations between CRF and overall as well as domain-specific physical literacy and intervention-induced changes in physical literacy among physically inactive adults.

This study was embedded in the project Primary Care und Physical Literacy as one out of two studies with a similar design and research questions [39]. The scoring procedure was adapted from the original questionnaire for the self-efficacy and physical activity behavior domains as well as for overall physical literacy. Comparing the outcomes of both studies, the results were in line for overall physical literacy and for the physical activity behavior, self-efficacy, and motivation domains, while improvements in the attitude and knowledge domains were only observed in the present study. Since participant characteristics were similar for both studies, these differences may be attributable to a higher adherence in the present study (68 vs. 48%). This in turn, may have been affected by conducting the intervention in spring and summer (vs. autumn and winter in the previous study) and by a stronger group cohesion among participants, as highlighted by the sport scientist who led the intervention in both studies. Indeed, both factors are known determinants of adherence in physical activity intervention studies [74,75]. Overall, a primary care setting was effective in the recruitment of physically inactive adults for the physical literacy-based intervention, especially when general practitioners were involved. Consequently, there should be an enhanced integration of primary care workers (i.e., health professionals) in prospective physical activity-promoting initiatives, as highlighted in the Austrian National Action Plan on physical activity [76].

There are some limitations that should be considered when interpreting the results of our study. First, we used a convenience sampling approach with self-selection for group allocation, which implies a potential risk of bias and affects the external validity. Second, since this was a pilot study, the sample size was relatively small and included mainly overweight middle-aged women, who showed a heterogeneity in adherence. Therefore, the present sample is not representative of the broader population of inactive adults. Future studies with larger samples, a balanced gender ratio, random allocation, and longer follow-ups are necessary. Third, even though we used valid and reliable questionnaires (subscales) to measure our five physical literacy domains, additional work is required to confirm the structural validity of our measurement instrument and the composite scoring procedure. The standardized composite scores were primarily used in the analyses for methodological reasons, rather than for feedback to the participants. In fact, this approach poses problems in terms of interpreting one’s physical literacy. Moreover, the IPAQ was used as a self-reported measurement of the participants’ physical activity behavior, which may overestimate the observed changes regarding this domain following the intervention [77]. Lastly, IG participants were allowed to attend either the strength-based, the multimodal-based, or both sessions on one intervention day. Consequently, it was not possible to investigate inter-group differences regarding the different types of sessions.

## 5. Conclusions

In conclusion, this study was designed to assess the association of a physical literacy-based intervention targeting physically inactive adults and changes in physical literacy. We demonstrated clear improvements in overall physical literacy and in four out of five physical literacy domains including physical activity behavior, attitude, knowledge, and self-efficacy in the IG. No changes were found for motivation by time and condition, which might be attributed to motivated participants at baseline resulting from self-selection. In addition, an inverse association between CRF at baseline and physical literacy changes was found. However, this relationship is tentative and should be interpreted with caution, with further research warranted. Overall, given the link between physical literacy and lifelong physical activity participation, our low-dose physical literacy-based intervention may be a promising approach to combat the high prevalence of physically inactive adults, not only in Austria, but also in other countries facing this challenge. Importantly, given the observed potential of primary care workers for the recruitment of physically inactive adults, they should be increasingly integrated into further physical activity-promoting initiatives.

## Figures and Tables

**Figure 1 ijerph-18-08593-f001:**
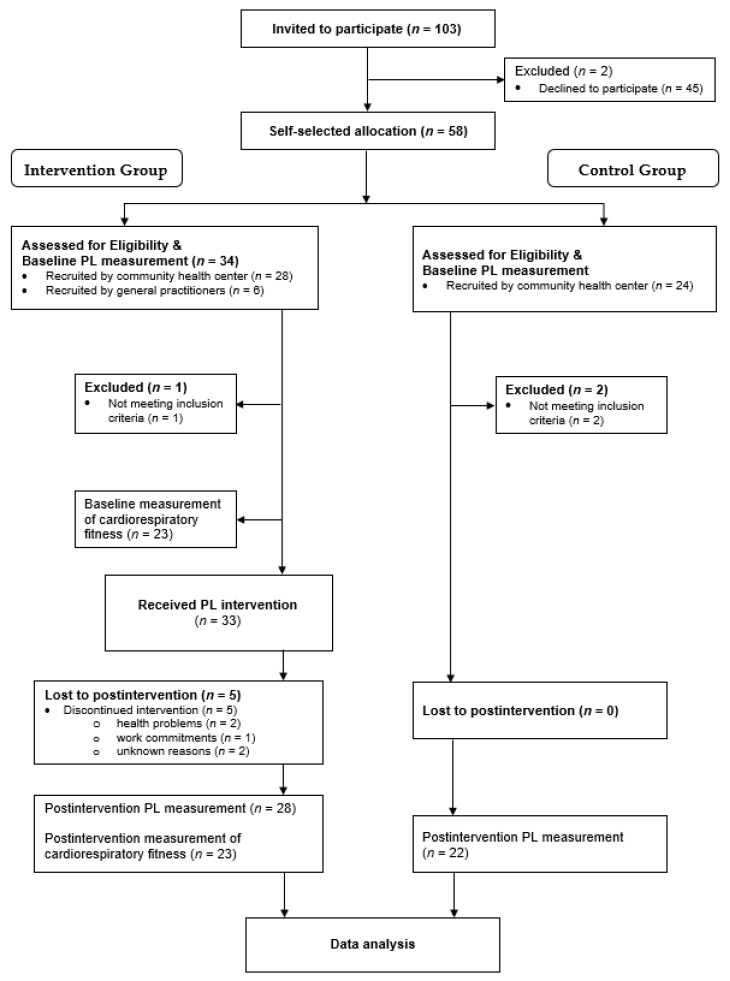
Flow diagram of recruitment numbers. PL: physical literacy.

**Figure 2 ijerph-18-08593-f002:**
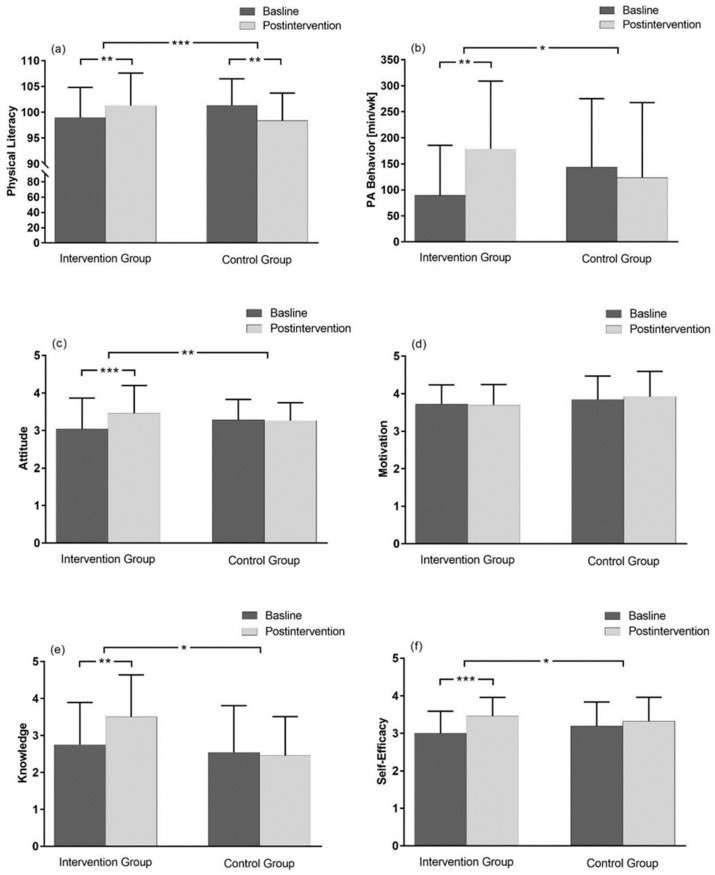
Physical literacy and all five domains of physical literacy by time and condition; (**a**): physical literacy, (**b**): physical activity behavior domain, (**c**): attitude/understanding domain, (**d**): motivation domain, (**e**): knowledge domain, (**f**): self-efficacy/confidence domain; Baseline: charcoal gray; postintervention: light gray; IG: intervention group; CG: control group. Values are given as mean ± SD; * *p* < 0.05; ** *p* < 0.01; *** *p* < 0.001.

**Table 1 ijerph-18-08593-t001:** Participant characteristics at baseline.

Characteristics	IG	CG	*p*-Value
**Gender (Female)**	25 (89%)	21 (96%)	0.43
**Age (Years)**	53 ± 10	50 ± 11	0.20
**Anthropometry**			
Height (cm)	166 ± 8	167 ± 5	0.69
Weight (kg)	77 ± 16	73 ± 11	0.35
BMI (kg/m²)	28 ± 5	26 ± 4	0.18
**Education Level *N* (%)**			
Compulsory school	0 (0%)	1 (4%)	0.46
Apprenticeship/Professional school degree	19 (68%)	11 (50%)	
School examination (A-Level)	5 (18%)	5 (23%)	
Grad	4 (14%)	5 (23%)	
**Chronic Diseases *N* (%)**	10 (36%)	3 (14%)	0.08
**Physical Literacy**	99.0 ± 5.8	101.3 ± 5.1	0.14
Physical activity behavior (min/wk)	89.6 ± 96.9	143.6 ± 131.5	0.10
Attitude/Understanding ^a^	3.0 ± 0.8	3.3 ± 0.6	0.25
Motivation ^a^	3.7 ± 0.5	3.8 ± 0.6	0.46
Knowledge ^a^	2.8 ± 1.1	2.6 ± 1.3	0.55
Self-efficacy/Confidence	3.0 ± 0.6	3.2 ± 0.6	0.26

Note: Values are given as mean ± SD or as frequency (%); IG: Intervention group; CG: control group; BMI: body-mass index; ^a^ values ranging between 1 and 5 with higher values indicating a greater proficiency.

## Data Availability

The data sets used and/or analyzed during the current study are available from the corresponding author upon reasonable request.

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
