# Peer review of "Primary Care and Physical Literacy: A Non-Randomized Controlled Pilot Study to Combat the High Prevalence of Physically Inactive Adults in Austria"

_ijerph, 2021, doi:10.3390/ijerph18168593_

Round 1

Reviewer 1 Report

Thank you for completing the revisions suggested. I find the paper's content and  transparency much improved. The inclusion of the supplementary files is especially welcomed.

Author Response

Thank you for evaluating this study and thus helping us to improve its quality and transparency.

Reviewer 2 Report

Thanks for the helpful responses and revisions.

Author Response

Thank you for evaluating this study and thus helping us to improve its quality and transparency. Now, we have had a professionsla English editing done, in order to improve the spelling and grammar throughout the manurscript. 

This manuscript is a resubmission of an earlier submission. The following is a list of the peer review reports and author responses from that submission.

Round 1

Reviewer 1 Report

Thank you for the opportunity to review this paper, which seek to assess the effects of a physical literacy intervention  in inactive adults. The authors have laid out a good rationale for studying the effects of physical literacy interventions. However, the study design is weak and therefore the conclusions that are drawn are questionable. It uses a very small sample size, with no sample size calculation and a non-randomised 'matched' design was used.

Here are my main comments on the paper:

Line 36: reference 4 is specifically for those with existing mental health condition - why highlight this reference?

Line 40: some readers may not know what specifically is recommended

Lines 47-48: This assumption is likely to lead to increased inequalities - I would suggest noting this.

Lines 49-51: The rationale linking physical literacy o long term physical activity behaviour is limited

Line 108: The aim infers that causal assumptions will be tested. This is causal tricky from non-rct that is not powered to detect changes.

Line 128: Baseline measure assessed by staff, but then described as self-assessment

Line 135: Schedules did not allow some to take part, therefore they were asked to take part in a control group. They were therefore not selected as a matched group - but a convenience sample? External validity of this data is very questionable

Line 148: This flow chart does not depict the recruitment process - those whose schedules did not meet the needs of the interventions were offered the control group - therefore common pool before this point, with an element of self selection. How many were approached but declined the invitation?

Line 163: Why was baseline cardiorespiratory fitness assessed in only 23/31 participants?

Line 200: It is not clear here that part of the paper is to  re-assess the psychometric properties of the amended scale? When I looked at the tool used in the original study, it was developed and tested in Children. Why chose to use this in an intervention study when it hasn't been tested in adults?

Line 207: The measurement scales should be summarised again

Line 227: Physical activity is measured with the self-reported IPAQ-SF, stating it is responsive to change. However the authors of one the the references state that "However one of these papers note: "caution is warranted when interpreting intervention change based solely on self-reported PA, as may overestimate change consistent with other research". This calls in to question the decisions made here.

Line 241: V02max is predicted maximal

Line 249: how were chronic diseases assessed

Line 297-312: The detailed statistical plan is welcome. However, the sample size is very very small. No sample size calculation is presented. I would imagine that the study is significantly underpowered for the different analytical questions included.

Line 467: The principal finding is stated as the intervention delivered large effects in PL. I do not believe such strong statements can be made from the design employed and the very small sample size. This has implications for the rest of the discussion

In addition, there are quite a number of spelling errors in the text: e.g.  line 20, 51, 151, 

Author Response

Thank you very much for sending us your very helpful comments on our manuscript. We feel that the changes suggested substantially improved our manuscript. Please find below our point-by-point response to your comments. A revised version of the manuscript is attached to this letter. We hope that the revised manuscript is now acceptable for publication.

Yours sincerely,

Peter Holler

Reviewer 2 Report

Thank you for giving me the chance to review the paper. I would agree with the gap identified for research examining adult PL interventions and commend the authors on their work in this space. Given the primary outcome is PL, I think there needs to be stronger rationale/reasoning for selecting what the authors have termed ‘domains’. As the team have adopted Whitehead’s 2013 definition (the rationale for this is unclear, it’s not the most current definition), the ‘domains’ presented in the current study do not align with Whitehead’s conceptualization of domains. I would suggest amending this term to avoid confusion.

Fundamentally, I’m unsure this questionnaire does give a Physical Literacy ‘level’. Although this is beyond the scope of this paper, those adopting a Whiteheadian approach to physical literacy would not advocate for the use of the term ‘levels’. I also wondered how, if any, information was provided back to participants? There are inconsistencies in the questionnaire in using PA/Exercise/Sport (this may be a translation issue).

The paper presents a potentially useful holistic intervention, involving primary care, but I had to go to the previously published article for further information regarding the intervention components. The trial registration should be included in this submission. For transparency and general good practice in intervention studies I would suggest including a TIDieR checklist in the supplementary material.

I have provided more detailed comments in the attached.

Author Response

(The authors gave the same response as above.)

Reviewer 3 Report

General comments

  1. This paper reports on a small trial examining effects of an intervention that targets elements of physical literacy. The idea is a good one and, given the sample of adults (rather than children) has an element of novelty.
  2. Overall, I agree that this kind of research is needed with adults. However, your sample is heavily skewed towards women. Did the results of the few men involved bias any of the findings? What would happen if you just analysed results for women and focused only on this?
  3. Do you need p values? Could you rely simply on effect sizes?
  4. Please consider a large reduction in the use of abbreviations. I recommend that physical literacy and physical activity are not abbreviated. Many journals are asking for this now. Later in the paper, constant referral to PL, PA, IG, CG, CRF etc gets tedious and confusing.  

Specific comments

  1. P1, Line (L) 41: population are putting
  2. P2, L48: I think this statement is too strong. I would tone down the statement, recognising other influences alongside individual responsibility.
  3. P2, L85: lack of time may be a commonly stated barrier, but it hides a lot. No research has shown that those with more time do more physical activity. It reflects people’s values and interests more than time per se. Perhaps a more nuanced comment could be made.
  4. P3, L105: consistency?
  5. P3, L115: delete second ‘was conducted’
  6. P4, Fig 1. I recommend this be re-drawn to better reflect CONSORT recommendations.
  7. P6: I wonder if a table might be helpful summarising the intervention elements in Section 2.5.
  8. P6, L284: ratio not ration
  9. P6, L293: a very high alpha value (i.e., >0.9) can also suggest item redundancy, so I would recommend not putting statements next to these values, other than >0.7 being acceptable.
  10. P7, baseline characteristics: to what extent did the few men affect these data?
  11. P8, Table 1. Add effect sizes? Also, the BMI data will be affected by gender. I wonder if you might be better off just including women.
  12. P10, L452: negatively
  13. P10, L462: a gateway to …
  14. P11, L493: physically
  15. P11, L523: adherence (delete ‘s’)
  16. P11, L524: delete ‘pointed out to be’ and replace with ‘was’
  17. P12, L525: when a (or the) general …
  18. P12, L532: future studies also need more men.

Author Response

Thank you very much for sending us your very helpful comments on our manuscript. We feel that the changes suggested improved our manuscript. Please find below our point-by-point response to your comments. A revised version of the manuscript is attached to this letter. We hope that the revised manuscript is now acceptable for publication.

Yours sincerely,

Peter Holler
